# Long-Term Studies of Wheat Leaf Rust in the North-Western Region of Russia

Elena Gultyaeva *[ID], Philipp Gannibal [ID] and Ekaterina Shaydayuk

All Russian Institute of Plant Protection, Shosse Podbelskogo 3, 196608 St. Petersburg, Russia
* Correspondence: egultyaeva@vizr.spb.ru

**Abstract:** Leaf rust, caused by *Puccinia triticina* (*Pt*), is a widely occurring disease of wheat in the North-Western Region of Russia. Annual *Pt* surveys of wheat in this region have been conducted between 2001 and 2021. In total, 740 single urediniospore isolates were analyzed over 20 years. Virulence to *Lr9*, *Lr19* and *Lr24* were rare in 2001–2010 and was not detected after 2010. Temporal variation in virulence was determined on Thatcher lines with *Lr1*, *Lr2a*, *Lr2b* and *Lr2c* genes and was found to be relatively high. Virulence to *Lr1* increased to 100% from 2001 to 2014. Until 2010, most northwestern *Pt* isolates were avirulent to *Lr2a* and virulent to *Lr2b* and *Lr2c*. In the middle of 2010, avirulence to *Lr2a*, *Lr2b*, *Lr2c* and *Lr15* began to increase. Strong variability between years was revealed for virulence to *Lr20* and *Lr26*. Based on a set of 20 differential lines, 122 virulence pathotypes were detected. More than half of those were observed only once across all years. Pathotypes were divided into groups of B-, C-, D- and F-, virulent to *Lr1* and *Lr2a*, dominating until 2009. From 2010 pathotype groups M- and P-, virulent to *Lr1* and avirulent to *Lr2a*, began to dominate. Temporal differentiation of northwestern *Pt* population for virulence was determined. High similarity was observed for *Pt* accessions in 2001–2009 and 2010–2015 and these two groups differed moderately from each other. *Pt* accessions from 2016–2019 and 2020–2021 differed from each other and from accessions from the previous collection period. Field response of *Lr* differential lines was studied in the North-Western Region during 1998–2022. Wheat genotypes with genes *Lr9*, *Lr19*, *Lr23*, *Lr24*, *Lr25*, *Lr28*, *Lr29*, *Lr35*, *Lr39*, *Lr42*, *Lr43*, *Lr45*, *Lr47*, *Lr48*, *Lr49*, *Lr50*, *Lr51*, *Lr53* and *Lr57* remained resistant throughout the period of the study. Leaf rust severity in lines Tc*Lr12*, Tc*Lr21*, Tc*Lr22a*, Gatcher (*Lr27+31*), Tc*Lr44* and Pavon (*Lr46*) varied from 1% to 30% before 2014 and significantly decreased after 2014. A general trend of decreasing virulence of the *Pt* pathogen has been observed in the North-Western Region over the recent years.

**Keywords:** *Lr* genes; population; *Triticum aestivum*; resistance; *Puccinia triticina*; virulence

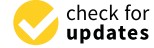



## 1. Introduction

Leaf rust, caused by *Puccinia triticina* Erikss. (*Pt*), occurs wherever wheat is grown and it is the most common and widely distributed of all cereal rusts. In Russia, its destructiveness has varied over time and between regions [1]. Until 2005, severe epidemics were reported in the European and Asian regions of Russia resulting in yield losses of 15% to 35% [2]. After 2010, the situation changed markedly with a decline in the occurrence of leaf rust evident in all regions of Russia and with no significant disease outbreaks reported.

Although fungicides can effectively control wheat rusts, growing resistant cultivars is a more efficient, economical and environmentally friendly and the optimal long-term strategy to minimize losses. In order to develop new cultivars with useful leaf rust resistance, it is necessary to introduce effective leaf rust resistance genes (*Lr*-genes) into new breeding material. Eighty-two leaf rust resistance genes are officially cataloged in wheat [3], but a limited number are used in breeding. Various *Lr* genes have been deployed in wheat cultivars in different Russian regions. The majority of Russian winter and spring wheat cultivars have less and partially effective *Lr* genes, *Lr1*, *Lr3*, *Lr10*, *Lr20*, *Lr26* and/or *Lr34*. Genes

*Lr9* and *Lr19* have mainly been incorporated into spring wheat cultivars and, accordingly, they are common in the areas of spring wheat cultivation (Volga, Ural and West Siberia). Winter cultivars, with *Lr9*, have been cultivated in the central European wheat-growing areas to a moderate extent. In 2015, cultivars with the adult plant resistance gene, *Lr37* began to grow in North Caucasus and, subsequently, (2020) in Central European regions. The first Russian spring cultivars with *Lr24* (Lider) and *Lr21* (Silantiy) were recommended for cultivation in West Siberia in 2020 and 2022, respectively. Spring cultivars with the highly effective *Lr6Agi1* and *Lr6Agi2* genes, having a substitution of wheat chromosome 6D with chromosomes 6Agi or 6Agi2, which is part of the J (=E) subgenome of *Thinopyrum intermedium*, are widely grown in Volga region. Spring cultivars with the effective gene *LrSp* transferred from *Aegilops speltoides* have been grown in Ural region since 2012 [4].

The leaf rust pathogen is highly variable with a moderate to large number of pathotypes (races) identified annually in the regional *Pt* populations. Changes in *Pt* population structure are mostly driven by the cultivars grown. Two leaf rust populations (European and the West Asian) have been recognized based on the long-time virulence surveys and SSR genotyping throughout the wheatbelt of Russia spreading from west to east across the Ural Mountains [5]. Most of regional *Pt* accessions from West Asian Russia were similar in virulence throughout a long-time study as a consequence of the genetic similarity (with a uniform set of resistance genes) of the wheat cultivars grown in this region. The similarity in virulence of European *Pt* accessions was somewhat lower. This is likely due to the differences in environmental conditions, types of wheat and a greater range of cultivars grown in that region. During long-time studies of *Pt* in European regions, in some years there was detectable differentiation of northwestern populations from other populations [6–8].

The North-Western Economic Region is located in the northern part of the Non-Chernozem zone of the Russian Federation, on the Russian (East European) plain. The region borders Latvia, Estonia, Belarus, Finland and the central regions of Russia. The region includes the Leningrad, Novgorod and Pskov regions and the federal city of St. Petersburg [9] (Figure 1). The North-Western Region is characterized by a temperate continental climate. More than 70% of agricultural production is animal husbandry, which is the defining industry of the region. Crop production is mostly focused on cereals, potatoes and forage crops, with winter and spring common wheat widely cultivated. Wheat is mainly grown as a fodder for the livestock industry. In recent years, wheat grown for grain propagation has increased substantially. In 2022, 99 kha of wheat was sown (38 kha winter wheat and 61 kha spring wheat) [10].

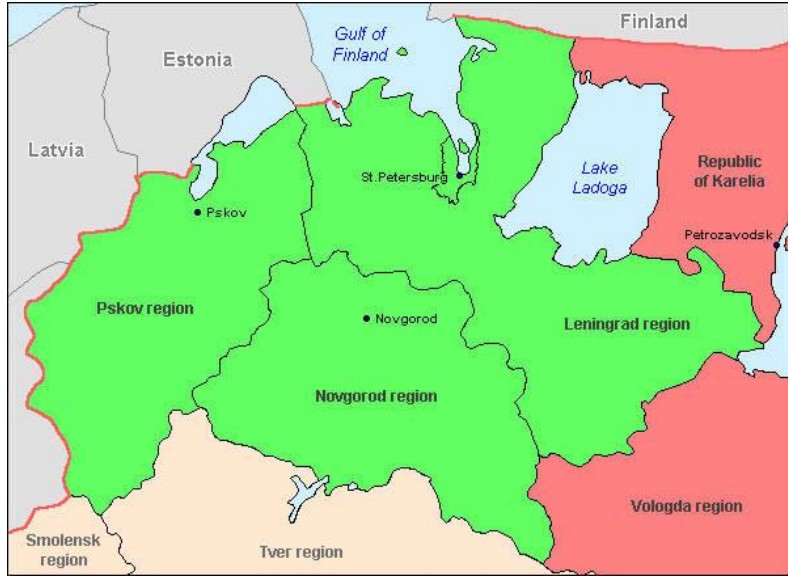

**Figure 1.** Location of the North-Western Region of Russia.

Climate change and a variety of unpredictable abiotic and biotic stresses have continually created threats to wheat production in this region. Among the biotic stresses, fungal diseases are the major constraints for wheat production. Leaf rust is the most common among the three wheat rusts in the North-Western Region. Under favorable conditions, a peak of leaf rust development may start in the beginning July and last until leaves senesce in August. Leaf rust severity varies from weak (1–5%) to high (50–80%). The importance of *Pt* in the North-Western Region, as in other Russian regions, was greatest up until the mid-2010s.

Currently, the disease is either absent or highly localized with minimal impact. A similar trend towards a decline in the leaf rust severity has also been observed in other regions of Russia [5] and all over the world [11,12]. However, there are a number studies on wheat leaf rust that predicted though simulation modeling an increasing risk of wheat leaf rust this century in the NW Europe [13–15]. Therefore, despite declining trends, it remains important to continue monitoring *Pt* populations.

*Pt* infection in the North-Western Region can be carried by wind from neighboring regions of Russia and Western Europe. This determines the high genetic diversity of the northwestern *Pt* population. Most of the wheat cultivars grown in this region before 2010 were highly susceptible to leaf rust (viz. cvs Mironovskaya 808, Inna, Moskovskaya 39, Leningradka, Irgina, Iren', Leningradskaya 97, Amir, MIS, Lada and Ester). With molecular markers, ineffective leaf rust resistance genes (*Lr*) were identified in some of these cultivars (e.g., *Lr10* in cvs Leningradka, Amir, MIS and Lada and *Lr1* in cv. Moskovskaya 39) [16]. From 2010, the range of commercial cultivars began to expand substantially. New domestic (cvs Galina, Skipetr, Moskovskaya 56 and Dar'ya) and foreign (cvs Skigen, Smuga, Balitus, Sonett, Calixo, Triso, Lad'ya and Sudarynya) cultivars begun to be cultivated in this region. No highly effective *Lr* genes have been identified in these cultivars, though many of them carry ineffective genes in combinations or individually. For example, genes *Lr10*, *Lr13* and *Lr17* have been identified in Skagen [17], *Lr10* and *Lr26* in Skipetr, *Lr1* and *Lr20* in Sonett, *Lr1* in Moskovskaya 56, *Lr20* in Dar'ya and Triso and *Lr10* in Leningradskaya 6 [16]. Changes in the genetic diversity of cultivated wheat cultivars could cause changes in the composition of northwestern *Pt* population.

*Pt* virulence surveys of wheat grown in the North-Western Region have been conducted annually by the All Russian Research Institute (St. Petersburg). These studies included virulence analysis of *Pt* populations at seedling and adult plant stages. This allowed the evaluation of the dynamics of the effectiveness of *Lr* genes, both as a function of time and pathotypes composition. The present contribution is a summary of annual virulence surveys of *Pt* of the northwestern *Pt* population at the seedling and adult plant stages from 2000 to present.

## 2. Materials and Methods

### 2.1. Puccinia triticina Virulence Analysis at the Seedling Stage

Accessions of *Pt* were obtained in 2001–2021 from a range of cultivars of common wheat growing in commercial fields, experimental plots, breeding and research nurseries in Leningrad, Pskov and Novgorod regions (Figure 1). *Pt* collection consisted of flag leaves with uredinial infections. One to 10 leaves of a single cultivar from each plot/field were used for one *Pt* sample. The leaves with uredinia were dried at room temperature, then refrigerated at 4 °C until processed for virulence characterization. Up to five single uredinial isolates of *Pt* were derived from each collection following the same procedures as previously described for *Pt* virulence surveys in Russia [5–7,18].

Single uredinial isolates of *Pt* were derived from each collection and were used to spray inoculate of 20 of 8–10 day-old differential Tc*Lr* lines. Pathotypes were determined 10–12 days after inoculation for each isolate using a 0–4 scale. Infection types (IT) 0–2+ (immune response to moderate uredinia with necrosis and/or chlorosis) were classified as avirulent and IT 3–4 (moderate to large uredinia without chlorosis or necrosis) were classified as virulent [19–21].

Each isolate was given a five letter code based on virulence/avirulence to each of the five sets of four differentials. North American race (pathotype) designation system uses a four letter code system to define the reaction on 20 of these Thatcher single *Lr* gene near-isogenic wheat lines. Sets 1 to 3 are similar to Long and Kolmer [19] (Set 1: *Lr1*, *Lr2a*, *Lr2c* and *Lr3a*; Set 2: *Lr9*, *Lr16*, *Lr24* and *Lr26*; and Set 3: *Lr3ka*, *Lr11*, *Lr17* and *Lr30*). Set 4 (*Lr2b*, *Lr3bg*, *Lr14a* and *Lr14b*) and Set 5 (*Lr15*, *Lr18*, *Lr19* and *Lr20*) were added to reveal polymorphisms in the Russian populations (Table 1) [5,22,23]. Thatcher was included as a susceptible control. Roger's distance was calculated for the evaluation of differences between annual *Pt* populations.

**Table 1.** Code for the 20 differential hosts for *Puccinia triticina* (*Pt*).

| *Pt* code [1] | Infection type produced on near isogenic *Lr* lines | | | |
|:---:|:---:|:---:|:---:|:---:|
| | Host set 1: *Lr1*, *Lr2a*, *Lr2c* and *Lr3a* Host set 2: *Lr9*, *Lr16*, *Lr24* and *Lr26* Host set 3: *Lr3ka*, *Lr11*, *Lr17* and *Lr30* Host set 4: *Lr2b*, *Lr3bg*, *Lr14a* and *Lr14b* Host set 5: *Lr15*, *Lr18*, *Lr19* and *Lr20* | | | |
| B | L | L | L | L |
| C | L | L | L | H |
| D | L | L | H | L |
| F | L | L | H | H |
| G | L | H | L | L |
| H | L | H | L | H |
| J | L | H | H | L |
| K | L | H | H | H |
| L | H | L | L | L |
| M | H | L | L | H |
| N | H | L | H | L |
| P | H | L | H | H |
| Q | H | H | L | L |
| R | H | H | L | H |
| S | H | H | H | L |
| T | H | H | H | H |

[1] *Pt* code consists of the designation for set 1, followed by that for set 2, etc. For example, race MGTFB: set 1 (M) -virulent to *Lr1*, *3a*; set 2 (G)—virulent to *Lrl6*; set 3 (T)—virulent, set 4 (F)—virulent to *Lr14a*, *Lr14b*; set 5 (B)—avirulent. L—low infection type (avirulent); H—high infection type (virulent).

## *2.2. Field Resistance Study*

An annual survey for leaf rust resistance was conducted in the field conditions in the North-Western Region of Russia. The studies were conducted in the experimental field of N. I. Vavilov All-Russian Institute of Plant Genetic Resources, Pushkin district of St Petersburg in 1998–2022. The plots were 1 m double rows each with three replicates. The Thatcher was planted after every 10 differential lines as susceptible check.

This evaluation used artificial inoculation with the local *Pt* populations and natural *Pt* infection. Natural infection was mostly used until 2010 when leaf rust caused a high severity of disease every year. However, since 2012, artificial inoculation of differentials in the field was needed. The field inoculation was conducted using a mixture of *Pt* isolates representing the most prevailing races in the North-Western region suspended in 0.03% Tween 20 onto the spreader rows at the tillering stage. The second inoculation was conducted after 10–12 days of no visible symptoms were observed. If weather conditions were

favorable for disease development, then high rust severity was observed. However, in some years (for example 2018 and 2021) the environmental conditions (high temperature and low humidity) were not conducive to disease development and only limited symptoms were evident.

A differential set included 52 Thatcher lines and wheat cultivars with *Lr*-genes were used in field test (Supplement Tables S1 and S2). The differential lines were rated for leaf rust severity using a 0–4 modified Cobb scale [24]. Additionally, they were rated for leaf rust response with the following scale: R (resistant), hypersensitive flecking with small necrotic uredinia (IT 0); MR (moderately resistant), small to moderate necrotic uredinia (IT 1-2); MS moderately susceptible), moderate to large uredinia with abundant chlorosis (IT 3; X); and S (susceptible), large uredinia lacking chlorosis (IT 4) [20].

## 3. Results

### 3.1. Puccinia triticina Response at the Seedling Stage

The determination of evolutionary trends in leaf rust races needs to be based on pathotype surveys. Such trends can only be determined from continuous observations made over many years. Virulence surveys aim to detect new pathotypes and monitor shifts of pathotype frequencies, and help breeders to develop efficient resistance strategies to this disease. Such virulence survey work has a long history in many wheat-producing areas in the world [25–32].

In this study, a total of 740 single urediniospore isolates from the North-West of Russia were analyzed over 20 years. The dynamic of *Pt* virulence frequencies is shown in Figure 2. Isolates virulent to *Lr9*, *Lr19* and *Lr24* were extremely rare and identified once in 2005 with frequency 4, 14 and 7%, respectively. The main changes in the northwestern *Pt* population were the frequencies of virulence to *Lr2a*, *Lr2b*, *Lr9*, *Lr15*, *Lr16*, *Lr20* and *Lr26*. Substantial variability in virulence between years was revealed in lines with *Lr1* gene. Virulence to *Lr1* was relatively low during 2001–2006 but reached extremely high levels after 2007. Since 2010, the same trend has been observed in other regions of Russia [5]. The virulence to *Lr2a* was higher in 2006–2008 (62–83%), but ranged from 0 to 46% in other years. The frequencies of isolates virulent to *Lr2b* and *Lr2c* genes have increased over the last 5 years and most were avirulent to *Lr15*. Avirulence to *Lr16* becomes predominant from 2019 onwards. High variability in virulence between years was revealed in lines with *Lr20* (41–100%) and *Lr26* (0–100%). Over the entire period (2001–2021), high virulence frequencies were observed against genes *Lr3a*, *Lr3bg*, *Lr3ka*, *Lr11*, *Lr14a*, *Lr14b*, *Lr17*, *Lr18* and *Lr30*.

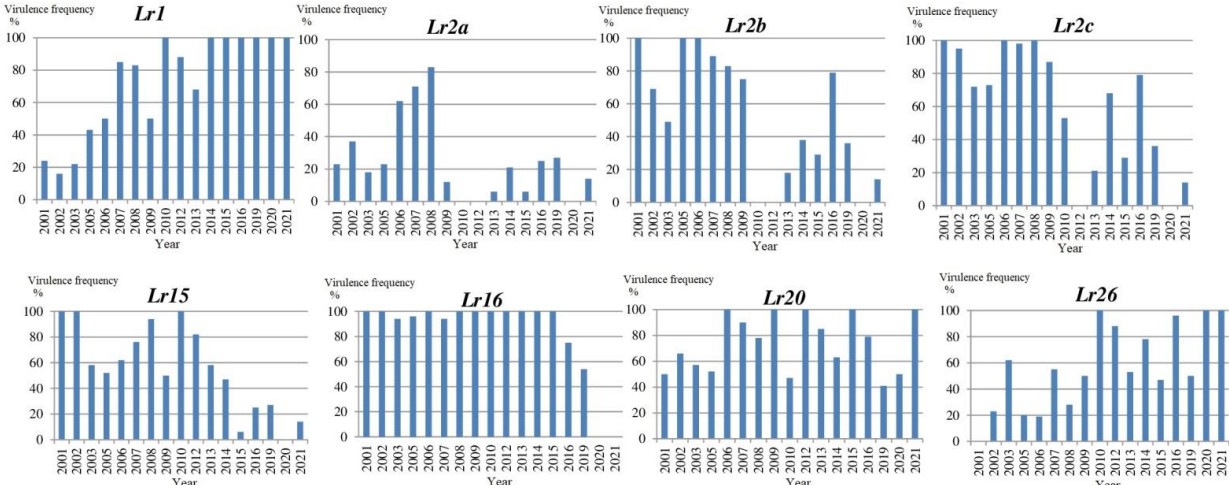

**Figure 2.** Frequencies of isolates of *Puccinia triticina* in North-Western Region of Russia in 2001–2021 virulent to Thatcher wheat lines with single genes for resistance (%).

Based on the set of 20 differential lines, 122 virulence pathotypes were detected among 740 isolates. Descriptive parameters of the *Pt* isolates are presented in Table 2. More than half of the pathotypes (72) were observed only once across all years. Rare *Pt* pathotypes were more frequent in accessions from the breeding nurseries or experimental fields. The virulence complexity (relative number of susceptible differential lines) of *Pt* accessions varied strongly between years. The highest virulence complexity was in *Pt* isolates collected from 2016, 2007 and 2008 (16.1–15.4) and the lowest in 2003, 2005, 2019, 2020 and 2021 (12.0–12.9). Kosman diversity within the annual populations was in the range 0 (2020) to 0.36 (2005).

**Table 2.** Descriptive parameters of northwestern population of *Puccinia triticina* in 2001–2021.

| Parameter | 2001 | 2002 | 2003 | 2005 | 2006 | 2007 | 2008 | 2009 | 2010 | 2012 | 2013 | 2014 | 2015 | 2016 | 2019 | 2020 | 2021 |
|---|---|---|---|---|---|---|---|---|---|---|---|---|---|---|---|---|---|
| Number of isolates | 34 | 56 | 67 | 44 | 16 | 139 | 36 | 8 | 17 | 17 | 132 | 79 | 36 | 24 | 22 | 6 | 7 |
| Number of pathotypes | 3 | 12 | 42 | 32 | 6 | 33 | 11 | 5 | 2 | 5 | 24 | 18 | 5 | 3 | 8 | 1 | 2 |
| Relative richness [1] | 0.1 | 0.21 | 0.63 | 0.73 | 0.37 | 0.24 | 0.31 | 0.62 | 0.12 | 0.29 | 0.18 | 0.23 | 0.14 | 0.12 | 0.36 | 0.17 | 0.28 |
| Abundance [2] (%) | 50 | 32 | 13 | 18 | 37 | 38 | 36 | 50 | 53 | 76 | 23 | 16 | 41 | 54 | 45 | 100 | 86 |
| Number of rare pathotypes [3] | 0 | 1 | 22 | 26 | 0 | 8 | 2 | 1 | 0 | 1 | 6 | 2 | 0 | 0 | 3 | 0 | 0 |
| Complexity [4] | 14 | 14.1 | 12.9 | 12.5 | 14.9 | 15.6 | 15.4 | 14.1 | 14 | 13.3 | 13.1 | 14.2 | 13.2 | 14 | 12.7 | 12 | 12.6 |
| Kosman diversity [5] | 0.1 | 0.15 | 0.29 | 0.36 | 0.14 | 0.14 | 0.10 | 0.21 | 0.09 | 0.07 | 0.18 | 0.2 | 0.12 | 0.16 | 0.25 | 0 | 0.06 |

[1] relative richness = number of pathotypes/number of isolates; [2] frequency of the predominant pathotype (%); [3] rare pathotype is that detected only once; [4] average value of the relative virulence complexity of isolates; [5] within a population diversity.

The main differences between virulence pathotypes were observed in the reaction type in lines with *Lr1* and *Lr2a*, in the first quadruple of the differential set. Significant changes were established in the frequencies of the following pathotype groups: F-, B- and C- (avirulence to *Lr1* and *Lr2a*); P- and M- (virulence to *Lr1* and avirulence to *Lr2a*); and T- (virulence to *Lr1* and *Lr2a*) (Tables 3 and 4). Avirulence to *Lr1* (pathotype groups B-, C-, D-, F-, H- and K-) dominated until 2009, which were highlighted in orange. Then, the frequencies of pathotypes virulent to *Lr1* and avirulent to *Lr2a* (pathotype groups M- and P-) increased to become predominant (highlighted in green). The frequencies of pathotypes virulent to both *Lr1* and *Lr2a* (T- group) varied between years. The highest frequencies of T-pathotypes were detected in accessions from 2007 and 2008 (83 and 67%, respectively) and they were not found in 2010, 2012 and 2020. The summers of 2010 and 2011 had unusually high temperatures during the wheat growing season (from late-June to mid-August), which could affect the change in pathotype composition of *Pt* populations.

The predominant pathotypes identified for two or more years with a frequency above 5% are presented in Table 4. No pathotype was common to all years. The group of closely related pathotypes, THTTR, TGTTR and TCTTR, was the most common over the duration of the study. These pathotypes differed in avirulence to *Lr16* (TCTTR) and virulence to *Lr26* (TGTTR). Pathotype THTTR was detected nine times over the 20 years and pathotype TGTTR seven times and were the predominant pathotypes until 2010. Pathotype TCTTR (avirulence to *Lr16*) becomes predominant from 2016 onwards. A similar situation was noted in France with virulence against *Lr16* found until 2011, but not afterwards [26].

Roger's distance was calculated for the evaluation of differences between annual *Pt* populations. It allows the comparison of two populations on the basis of the frequencies of pathotypes that occur in the populations [32]. High similarity in pathotypes composition was detected in 2001–2009 and 2010–2015, but the composition differed moderately between these periods (Figure 3). Accessions of *Pt* from 2016 and later differed from those periods.

**Table 3.** Frequencies of pathotype groups of *Puccinia triticina* in North-Western Region of Russia in 2001–2021.

| Year | Pathotype Group Frequency (%) | | | | | | | | | | | |
|---|---|---|---|---|---|---|---|---|---|---|---|---|
| | B- | C- | D- | F- | K- | L- | M- | N- | P- | Q- | R- | T- |
| 2001 | 0 | 0 | 0 | 77 | 0 | 0 | 0 | 0 | 0 | 0 | 0 | 23 |
| 2002 | 0 | 6 | 0 | 56 | 22 | 0 | 0 | 0 | 0 | 0 | 0 | 16 |
| 2003 | 0 | 19 | 0 | 50 | 0 | 0 | 1 | 0 | 4 | 1 | 3 | 22 |
| 2005 | 17 | 2 | 4 | 19 | 0 | 4 | 2 | 0 | 13 | 0 | 0 | 39 |
| 2006 | 0 | 0 | 0 | 25 | 25 | 0 | 0 | 0 | 12 | 0 | 0 | 38 |
| 2007 | 0 | 0 | 0 | 13 | 3 | 0 | 3 | 0 | 14 | 0 | 0 | 67 |
| 2008 | 0 | 0 | 0 | 17 | 0 | 0 | 0 | 0 | 0 | 0 | 0 | 83 |
| 2009 | 0 | 0 | 0 | 50 | 0 | 0 | 12 | 0 | 25 | 0 | 0 | 13 |
| 2010 | 0 | 0 | 0 | 0 | 0 | 0 | 47 | 0 | 53 | 0 | 0 | 0 |
| 2012 | 6 | 6 | 0 | 0 | 0 | 0 | 88 | 0 | 0 | 0 | 0 | 0 |
| 2013 | 0 | 0 | 0 | 1 | 0 | 2 | 45 | 0 | 15 | 0 | 0 | 37 |
| 2014 | 0 | 0 | 0 | 0 | 0 | 1 | 30 | 1 | 46 | 0 | 0 | 22 |
| 2015 | 0 | 0 | 0 | 0 | 0 | 0 | 70 | 0 | 23 | 0 | 0 | 7 |
| 2016 | 0 | 0 | 0 | 0 | 0 | 0 | 0 | 0 | 54 | 0 | 0 | 46 |
| 2019 | 0 | 0 | 0 | 0 | 0 | 0 | 51 | 0 | 10 | 0 | 0 | 39 |
| 2020 | 0 | 0 | 0 | 0 | 0 | 0 | 100 | 0 | 0 | 0 | 0 | 0 |
| 2021 | 0 | 0 | 0 | 0 | 0 | 0 | 86 | 0 | 0 | 0 | 0 | 14 |

Pathotypes avirulent to *Lr1* is highlighted in orange, pathotypes virulent to *Lr1* and avirulent to *Lr2a* is highlighted in green.

**Table 4.** The predominant pathotypes of *Puccinia triticina* and their frequencies in North-Western Region of Russia in 2001–2021.

| Pathotype | Avirulence *Lr*-Gene | 2001 | 2002 | 2003 | 2005 | 2006 | 2007 | 2008 | 2009 | 2010 | 2012 | 2013 | 2014 | 2015 | 2016 | 2019 | 2020 | 2021 |
|---|---|---|---|---|---|---|---|---|---|---|---|---|---|---|---|---|---|---|
| FGTTH | 1, 2a, 9, 15, 19, 24, 26 | 0 | 0 | 3 | 0 | 0 | 4 | 3 | 50 | 0 | 0 | 0 | 0 | 0 | 0 | 0 | 0 | 0 |
| FGTTR | 1, 2a, 9, 19, 24, 26 | 26 | 32 | 0 | 0 | 25 | 3 | 0 | 0 | 0 | 0 | 0 | 0 | 0 | 0 | 0 | 0 | 0 |
| KGTTR | 1, 9, 19, 24, 26 | 0 | 5 | 0 | 0 | 12 | 2 | 0 | 0 | 0 | 0 | 0 | 0 | 0 | 0 | 0 | 0 | 0 |
| KHTTR | 1, 9, 19, 24 | 0 | 2 | 0 | 0 | 12 | 1 | 0 | 0 | 0 | 0 | 0 | 0 | 0 | 0 | 0 | 0 | 0 |
| MCTKH | 2a, 2b, 2c, 9, 15, 16 19, 24 | 0 | 0 | 0 | 0 | 0 | 1 | 0 | 0 | 0 | 0 | 0 | 0 | 0 | 0 | 4 | 100 | 86 |
| MHTKH | 2a, 2b, 2c, 9, 15, 19, 24 | 0 | 0 | 0 | 0 | 0 | 0 | 0 | 0 | 0 | 6 | 3 | 6 | 41 | 0 | 0 | 0 | 0 |
| MGTKG | 2a, 2b, 2c, 9, 15, 19, 20, 24, 26 | 0 | 0 | 0 | 0 | 0 | 1 | 0 | 0 | 0 | 0 | 0 | 0 | 0 | 21 | 45 | 0 | 0 |
| MGTKH | 2a, 2b, 2c, 9, 15, 19, 24, 26 | 0 | 0 | 0 | 0 | 0 | 1 | 0 | 0 | 0 | 0 | 8 | 0 | 29 | 0 | 0 | 0 | 0 |
| MHTKQ | 2a, 2b, 2c, 9, 19, 20, 24 | 0 | 0 | 0 | 0 | 0 | 0 | 0 | 0 | 0 | 0 | 9 | 8 | 0 | 0 | 0 | 0 | 0 |
| MHTKR | 2a, 2b, 2c, 9, 19, 24 | 0 | 0 | 0 | 0 | 0 | 0 | 0 | 0 | 47 | 76 | 21 | 16 | 0 | 0 | 0 | 0 | 0 |
| PGTTH | 2a, 9, 15, 19, 24, 26 | 0 | 0 | 0 | 2 | 0 | 4 | 0 | 0 | 0 | 0 | 0 | 2 | 18 | 0 | 0 | 0 | 0 |
| PHTTH | 2a, 9, 15, 19, 24 | 0 | 0 | 0 | 0 | 0 | 0 | 0 | 0 | 0 | 0 | 1 | 8 | 6 | 54 | 0 | 0 | 0 |
| PGTTR | 2a, 9, 19, 24, 26 | 0 | 0 | 0 | 0 | 6 | 1 | 0 | 0 | 0 | 0 | 3 | 0 | 0 | 0 | 0 | 0 | 0 |
| PHTTR | 2a, 9, 19, 24 | 0 | 0 | 0 | 0 | 6 | 1 | 0 | 12 | 0 | 0 | 7 | 1 | 0 | 0 | 0 | 0 | 0 |
| PHTKR | 2a, 2b, 9, 19, 24 | 0 | 0 | 0 | 0 | 0 | 2 | 0 | 12 | 0 | 0 | 1 | 0 | 0 | 0 | 0 | 0 | 0 |
| TCTTR | 9, 16, 19, 24 | 0 | 0 | 0 | 0 | 0 | 1 | 0 | 0 | 0 | 0 | 0 | 0 | 0 | 25 | 23 | 0 | 14 |
| TGTTR | 9, 19, 24, 26 | 23 | 0 | 3 | 0 | 0 | 15 | 36 | 0 | 0 | 0 | 4 | 9 | 6 | 0 | 0 | 0 | 0 |
| THTTR | 9, 19, 24 | 0 | 14 | 6 | 18 | 0 | 38 | 17 | 12 | 0 | 0 | 2 | 1 | 0 | 0 | 4 | 0 | 0 |
| TGTTH | 9, 15, 19, 20, 24, 26 | 0 | 0 | 0 | 2 | 37 | 2 | 0 | 0 | 0 | 0 | 0 | 0 | 0 | 0 | 0 | 0 | 0 |
| TGTTQ | 9, 19, 20, 24, 26 | 0 | 0 | 0 | 0 | 0 | 1 | 19 | 0 | 0 | 0 | 4 | 0 | 0 | 0 | 0 | 0 | 0 |
| THTTQ | 9, 19, 20, 24 | 0 | 0 | 1 | 0 | 0 | 3 | 3 | 0 | 0 | 0 | 0 | 6 | 0 | 0 | 0 | 0 | 0 |

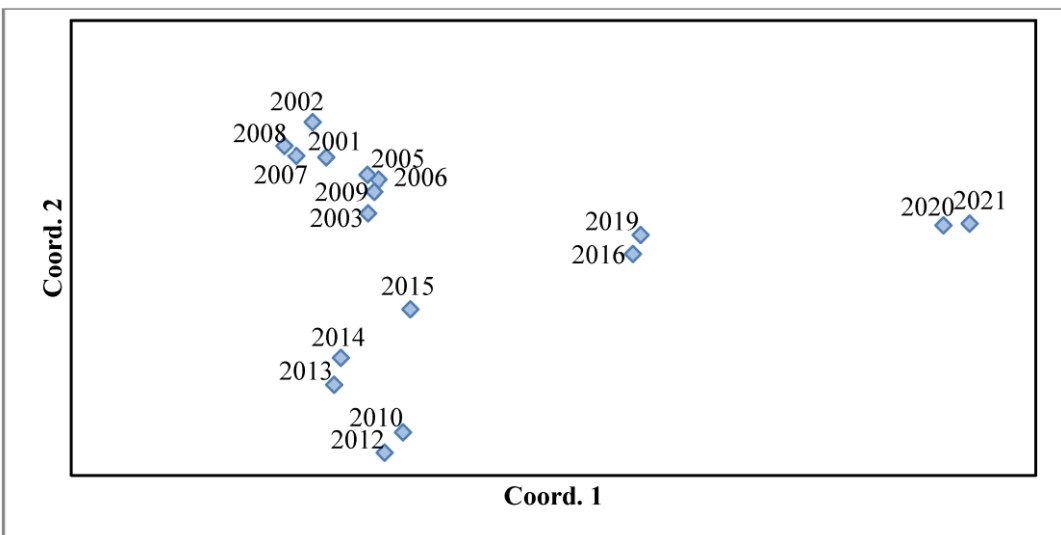

**Figure 3.** Principal component analysis of relationships among northwestern accessions of *Puccinia triticina* on common wheat in 2001–2021 based on the Roger's distance.

### 3.2. Lr Gene Effectiveness in the Field

Field testing provides an additional approach to survey the occurrence of virulence. Genetic resistance to rust pathogens is usually divided into three general categories [33,34]: (1) race-specific seedling resistance; (2) race-specific adult plant resistance (APR); and (3) race non-specific APR, also known as slow-rusting or partial resistance. Race-specific resistance, seedling resistance or all-stage resistance are often characterized by a strong to moderate resistance reaction usually associated with the hypersensitive response that prevents fungal infection and sporulation at all developmental stages when the pathogen possesses a corresponding avirulence gene. Most of catalogued *Lr* genes are race-specific. Race non-specific APR usually gives a susceptible response at the seedling stage and is expressed quantitatively at post-seedling growth stages either as slow rusting or partial resistance. It is usually characterized by lower frequencies of infection, longer latency period, smaller uredinia and reduced urediniospore production [34]. Only a few genes, for example, *Lr34/Yr18/Pm38/Sr57*, *Lr46/Yr29/Pm39/Sr58*, *Lr67/Yr46/Pm46/Sr55* and *Lr68*, are known as slow rusting or partial resistance [35–40]. The effects of these genes, when present, alone are moderate, nevertheless their contribution is useful in gene combinations and interactions with other major genes and a range of minor QTLs that cause additive effects, resulting in high levels of durable resistance [37]. The final type is race-specific APR, which includes *Lr12*, *Lr13*, *Lr22a*, *Lr35*, *Lr37*, *Lr48*, *Lr49*, *Lr74*, *Lr75* and *Lr77* [3,41,42]. Most of them confer hypersensitive reactions and interact with pathogen in a gene-for-gene manner.

Field testing provides the majority of information on the effectiveness of adult plant resistance genes and seedling resistance genes. Wheat lines and cultivars with seedling resistance genes *Lr9*, *Lr19*, *Lr24*, *Lr29*, *Lr41*, *Lr42*, *Lr43*, *Lr45*, *Lr47*, *Lr51*, *Lr53* and *Lr57* were highly resistant during long-time study (Supplement Tables S1 and S2). Line Tc*Lr28* was weakly affected twice over the duration of the study (type reaction MR in 2004 with 10% severity and in 2005 with 3% severity) and likewise for line KS96WGRC36 (*Lr50*) (MR in 2007 and 2016 with 1% severity) and line Tc*Lr25* (S in 2018 with 1% severity). Low rust severity (up to 10%) was detected in lines Tc*Lr23*, Tc*Lr35*, CSP44 (*Lr48*), VL404.№8677 (*Lr49*), KS86WGRC02 (*Lr39*), KS89WGRC07 *Lr40* (=21) and KS96WGRC36 (*Lr50*). Leaf rust severity in lines Tc*Lr12*, Tc*Lr21*, Tc*Lr22a*, Gatcher (*Lr27+31*), Tc*Lr44* and Pavon (*Lr46*) varied from 1% to 30% before 2014, but subsequently decreased substantially. Significant variation in disease severity (from 0% to 80%) were detected in lines with genes *Lr11*, *Lr13*, *Lr18*, *Lr27+31*, *Lr32*, *Lr52* and *Lr64*.

Variation of rust severity on lines with ARP genes is presented in Figure 4.

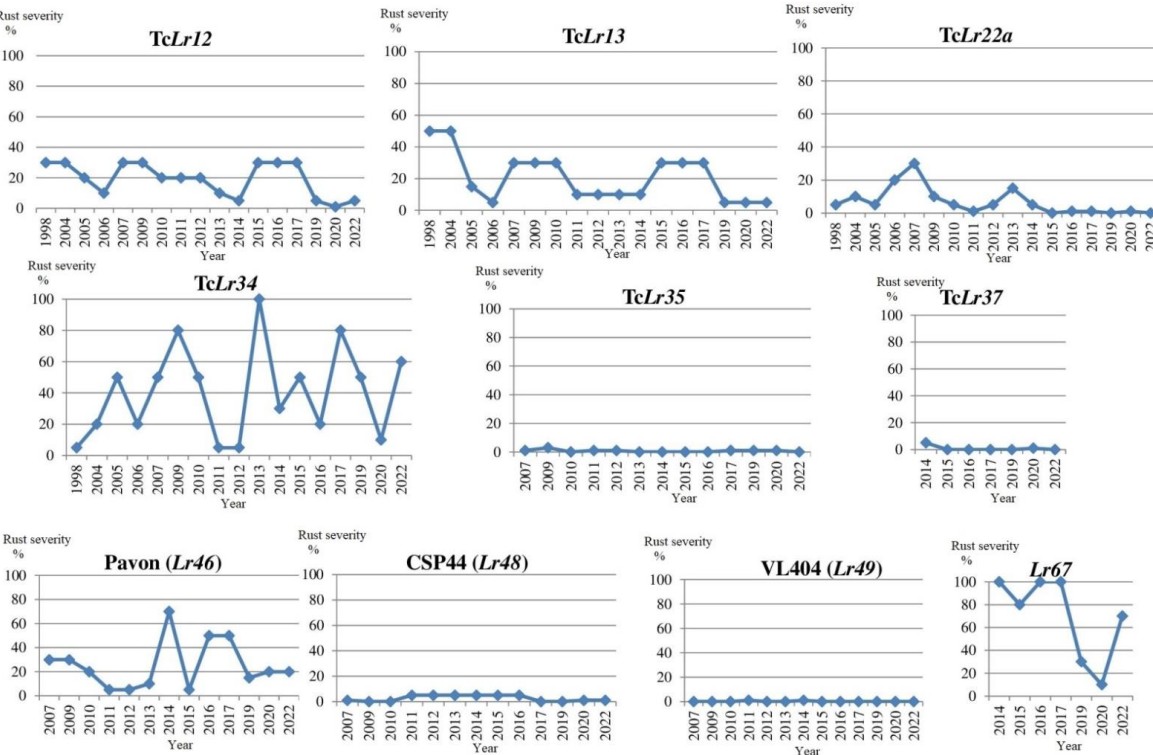

**Figure 4.** Leaf rust severity on Thatcher lines and cultivars with adult plant resistance genes in the North-Western region of Russia in 1998–2022.

## 4. Discussion

Wheat leaf rust has been one of the most damaging foliar diseases of wheat worldwide and it has consequently been a major focus of research, both in internationally [27,30], http://rustwatch.au.dk (accessed on 15 December 2022) and regionally [20,25,26,28–32]. *Pt* populations are characterized by high diversity, as a consequence of a combination of sexual and asexual reproduction, mutation and long-distance dispersal of urediniospores between wheat-growing areas [26]. Virulence surveys aim to detect new pathotypes, monitor shifts of pathotype frequencies and help breeders in proposing efficient resistance strategies for disease management [27].

The 2001–2021 survey gives a comprehensive picture of the virulence frequencies of the wheat leaf rust fungus populations in the North-Western region. A high variability in pathogen virulence was revealed. Generally, changes in the frequencies of pathotypes in the *Pt* populations depend on the changes and prevalence of *Lr* genes in the spectrum of wheat genotypes grown [26]. However, this does not apply to the North-Western Region, where no cultivars with effective single genes are grown. It appeared that this did not lead to the emergence of new virulence to highly effective genes, with the main changes of the northwestern population were driven by changes in the frequencies of ineffective genes. Thus, the susceptibility of widely-grown cultivars and the absence of highly effective *Lr* genes underpinned the selection of less virulent *Pt* pathotypes to the genotypes grown.

Genes *Lr9*, *Lr19*, *Lr24* and *Lr28* were highly effective against leaf rust in the North-Western Region at the seedling and adult plant stages. At the adult plant stage selection of these resistant lines was done on the basis of their reactions in small plots in the experimental fields. When tested in this manner, a cultivar may remain highly resistant for many years when selection pressure is negligible under these conditions. However, if a cultivar with this type of resistance becomes widely grown, selection pressure is high and races in the population with corresponding virulence may increase rapidly. An example of this is the loss of effectiveness of *Lr19* and *Lr9* in some Russian regions where cultivars with these genes have been widely grown. The first virulence to *Lr19* was found in the mid-1990s

in Volga region [43] and to *Lr9* in 2007 in Ural and West Siberian regions [44]. Until 2010, virulence to *Lr19* was present in *Pt* populations in many Russian wheat growing regions, including the North-Western region. The frequency of virulence to *TcLr19* was high before 2010, but subsequently declined, whereas virulence to *Lr9* increased annually during this period [4]. However, these genes have remained highly effective in the last decade in the central European regions and the North Caucasus [5].

Effectiveness of genes *Lr9*, *Lr19*, *Lr24* and *Lr28* varies in western European countries. Among them, *Lr9* and *Lr19* have been the most effective [26,28,29]. Low frequencies of virulence were recorded for *Lr24* in the Czech in 2004–2011, but, subsequently, it rarely exceeded 10%. Virulence to *Lr28* has slightly increased since 2015 due to the large growing area of cultivars possessing *Lr28* [28,29]. In France, virulence to *Lr24* and *Lr28* appeared in 2011, which, in the case of *Lr28*, commenced with the registration of cultivars carrying *Lr28*, but began before the widespread deployment of these cultivars [26]. These examples indicate that if cultivars with genes *Lr24* and *Lr28* were to be grown in the North-Western Region of Russia, it could lead to a loss of effectiveness of these genes due to wind dispersal of *Pt* urediniospores.

Fontyn et al. [26] analyzed the relationship between the *Lr* genes in the cultivars and virulence in the pathotypes in France in 2006–2016. It was shown that a small number of pathotypes dominated in French *Pt* populations. The frequency of the most frequent pathotypes correlated with the distribution of cultivated varieties and prevalence of *Lr* genes. However, certain pathotype–cultivar associations could not be explained solely by the distribution of *Lr* genes. Some virulences were detected in *Pt* populations, despite the absence of corresponding *Lr* genes in growing varieties [26].

The main changes of *Pt* population in the North-Western Region were apparently driven by changes in virulence frequencies to *Lr1*, *Lr2a*, *Lr2b* and *Lr2c* genes. Virulence to *Lr1* increased to 100% from 2001 to 2014. A similar trend in virulence to *Lr1* was found in all European Russian regions since 2010, with nearly all European isolates virulent on *Lr1* in 2012–2018 [5].

Such situation also was observed in the Czech Republic. There was a low average virulence frequency for *Lr1* during 2002–2007, followed by a rapid increase from 2008, reaching 100 and 98% in 2013 and 2018, respectively [28,29]. Another change in northwestern *Pt* population was an increase in isolates avirulent to *Lr2a*, *Lr2b* and *Lr2c*. Until 2010, most northwestern *Pt* isolates were avirulent to *Lr2a*, and virulent to *Lr2b* and *Lr2c*, then the frequency of isolates avirulent to three alleles of gene *Lr2* began to increase. During that period, a similar trend for these genes was observed in the Czech Republic. Additionally, in both countries, a high frequency of virulence to *Lr3*, *Lr10*, *Lr17* and *Lr26* was found [28,29].

Predominant pathotypes detected in the North-Western Region over the different time periods in this study differed mainly in virulence to *Lr1*, *Lr2a*, *Lr2b*, *Lr2c* and *Lr16*. Most of these pathotypes were widely distributed in other regions of Russia [5] and around the world [30]. The general trend of decreasing virulence of the *Pt* pathogen has been observed in the North-Western Region over the last five years.

In this study, there was good general agreement between the field tests and the virulence surveys. Wheat lines with genes *Lr9*, *Lr19*, *Lr24*, *Lr28*, *Lr29*, *Lr41*, *Lr42*, *Lr43*, *Lr45*, *Lr47*, *Lr51*, *Lr53* and *Lr57* remained free of infection in the North-Western region and no virulence was found. The lines Tc*Lr1*, Tc*Lr2a*, Tc*Lr2b*, Tc*Lr2c*,Tc*Lr3a*, Tc*Lr3bg*, Tc*Lr3ka*, Tc*Lr10*, Tc*Lr14a*, Tc*Lr15*, Tc*Lr20*, Tc*Lr26* and Tc*Lr30* had high susceptibility in the field and virulence frequencies for these genes at the seedling stage was also significant.

## 5. Conclusions

In this study, we analyzed the evolution of the *Pt* population in the North-Western Region of Russia over the last 20 years. The dynamics of virulence frequency and pathotype composition at the seedling stage and effectiveness of *Lr*-genes in the field were characterized. The temporal changes of the northwestern *Pt* population were determined between

2000–2010, 2011–2015 and 2016–2021. A general trend of decreasing virulence of the *Pt* was observed over the last 5 years.

Varying impact of disease was observed during this period. Leaf rust caused significant crop damage until 2012, but then declined in significance. In recent years, there has been a decline in the development of leaf rust pathogen on wheat worldwide [11]. This is likely to be due to a combination of climate change, reduced tillage, deployment of resistant cultivars and increased use in fungicides. Reduced tillage led to the emergence of other diseases, such as leaf spots, root rots and fusarium head blight. Consequently, stripe rust is now the most damaging and prevalent cereal rust in Western Europe and worldwide and in the North-Western Region of Russia. However, it should not be assumed that previously widespread diseases, including leaf rust, would not reemerge to pose a substantive threat to global wheat production. Rust spores can be wind dispersed over long distances, which can contribute to the rapid spread of new virulent pathotypes. Therefore, despite recent declines in leaf rust impacts, it is imperative to continue monitor the increased virulence in pathogen populations to ensure timely responses can be provided.

**Supplementary Materials:** The following supporting information can be downloaded at: https://www.mdpi.com/article/10.3390/agriculture13020255/s1, Table S1: Leaf rust severity in the field on differential lines in the North-Western Region of Russia in 1998–2013; Table S2: Leaf rust severity in the field on differential lines in the North-West of Russia in 2014–2022.

**Author Contributions:** Conceptualization and data analysis design, E.G.; methodology and performing experiments, E.G. and E.S.; data analysis, E.G.; interpretation of results, E.G. and E.S; data acquisition and curation, E.G. and P.G.; drafting the manuscript, E.G. All authors have read and agreed to the published version of the manuscript.

**Funding:** This research was carried out in the framework of the state task delegated to the All-Russian Research Institute of Plant Protection, Project FGEU-2022-0003 "Taxonomic, genetic and ecological diversity of the most important groups of phytopathogenic fungi".

**Institutional Review Board Statement:** Not applicable.

**Data Availability Statement:** All data are provided in the manuscript.

**Acknowledgments:** The editorial support of Ian Riley is highly appreciated.

**Conflicts of Interest:** The authors declare no conflict of interest.

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
