# Peer review of "Long-Term Studies of Wheat Leaf Rust in the North-Western Region of Russia"

_agriculture, doi:10.3390/agriculture13020255_

Round 1
Reviewer 1 Report
line 191-192, fig. 3:
from the caption it is not possible to understand the content of the graph shown.
The description of the Y-axis in each graph is missing, please make it.
The structure of the graphics is not uniform and therefore not acceptable. Sometimes only one isoalt is shown, sometimes more.
The presentation is not acceptable and should be changed by the author.
Line 208. tab. 2:
The criteria used to classify the pathotype groups (e.g. B-, C-, or K-) are not described. There is no indication of literary sources for this.
Please make changes and add them under Material and Methods.
Line 248: why is the race specific seedling resistance described as all stage resistance, it could be different.
Please ask the author to explain.
Line 283; Fig. 4: in all four subgraphs the labelling of the axes is missing. The facts cannot be understood from the caption alone.
The illustrations are not acceptable in their present form; please have the authors make changes.
line 286, discussion:
in the discussion, only the results are repeated in some detail. the intensive discussion with other international publications (e.g. M. Hovmöller, DK) is missing. The importance of the change in the occurrence is also discussed. The importance of the change in the occurrence of certain virulences is also elaborated.
What is the significance of certain resistance genes in the varieties or certain virulence genes in this airborne pathogen?
The authors are asked to work intensively on the discussion.
Author Response
Dear reviewer,
Thank you for your positive assessment of our work.
Answers to comments.
- line 191-192, fig. 3: from the caption it is not possible to understand the content of the graph shown.The description of the Y-axis in each graph is missing, please make it. The structure of the graphics is not uniform and therefore not acceptable. Sometimes only one isoalt is shown, sometimes more. The presentation is not acceptable and should be changed by the author.
Corrected. The figure has been changed in accordance with the reviewer's recommendation.
- Line 208. tab. 2: The criteria used to classify the pathotype groups (e.g. B-, C-, or K-) are not described. There is no indication of literary sources for this. Please make changes and add them under Material and Methods.
Corrected. A table with the classification of phenotypes has been added to the section "Material and methods".
- Line 248: why is the race specific seedling resistance described as all stage resistance, it could be different. Please ask the author to explain.
This information was provided in accordance with the cited sources. We have deleted phrase «as all stage resistance».
- Line 283; Fig. 4: in all four subgraphs the labelling of the axes is missing. The facts cannot be understood from the caption alone. The illustrations are not acceptable in their present form; please have the authors make changes.
Corrected.
- line 286, discussion: in the discussion, only the results are repeated in some detail. the intensive discussion with other international publications (e.g. M. Hovmöller, DK) is missing. The importance of the change in the occurrence is also discussed. The importance of the change in the occurrence of certain virulences is also elaborated. What is the significance of certain resistance genes in the varieties or certain virulence genes in this airborne pathogen? The authors are asked to work intensively on the discussion.
We do not agree that in the discussion, only the results are released in some detail. We compared our results with those published by other authors ( Fontyn et al., 2022; Mesterhazy et al., 2000; Hanzalová et al., 2020,2021; Huerta-Espino et al., 2011; Kolmer et al., 2015; McCallum et al., 2016). In Discussion, we mainly tried to compare the Pt population from North-West with the population from Western Europe, because the North-Western region of Russia is located close to them.
But, unfortunately, over the past five years there have been very few publications about the European wheat leaf rust populations and we included all of them. Also we compared our result with other international publications from previous time (e.g. Mesterhazy et al., 2000; Huerta-Espino et al., 2011). We could not include the publications of M. Hovmöller from Aarhus University because in the last decades the most of his works connected with yellow and stem rust. His works are widely discussed by us in articles about yellow rust, but in this one we do not see the need to present them.
The significance of certain resistance genes in the varieties or certain virulence genes in this airborne pathogen also was also discussed in the previous version of the discussion (e.g, paragraphs 3 and 4). But according to the suggestion of the reviewer, we are adding some new information in the discussion.
Thank you very much for the comments and suggestions to improve the manuscript.
With kind regards,
Elena Gultyaeva
Reviewer 2 Report
The manuscript " LONG-TERM STUDIES OF WHEAT LEAF RUST IN THE NORTH-WESTERN REGION OF RUSSIA" describes a long term detection for virulence of Puccinia triticina, which is very important for the disease control and useful to wheat breeding programs. In general, it is well addressed for the virulence dynamics and evolution, especially field pathotype essay is also carried out which is unuasul in other similiar researches. There was still some suggestions or questions as follwings:
1. Line 7, is widely occurring-->is a widely occurring;
2. Line 10, Lr19 should be italic;
3. Line 16, More than half of there-->More than half of these or those?
4. Line 32, Eriks.-->Erikss.
5. Line 45, The majority of in -->The majority of
6. Line 126, growing in in -->growing in
7. Line 137, tomoderate --> to moderate
8. Line 178, Figure 2 might be misordered, please check it and renew the according figures' order. I could not find the fig.2
8. Line 196, across al years -->across all years
9. Line 289, rustw atch.au.dk-->rustwatch.au.dk, delete space
10. Line 297, was reveal --> was revealed
11. Line 298, "depend on the changes in and " should be rethough about the usage.
12. Line 367, stripe rust in now the --> stripe rust is now the
13. When you did field inoculations with Pt race, how many you inoculate for each year and how to invoid contaminations between isolates? The detailed procedure were recommended to add. Since it is important for avoiding contaminations when phenotyping in a open field. And how to avoid contaminations?
14. Line 166-167, you are using the modified Cobb scale for evaluation of leaf rust resistance, the scale MS (which score 2-3) probably need to rethougt, since normally scale 2 is recorded in MR in most cases. Even though that in the field cases, the phenotype is very complicated, 2-3 was recorded MS might misleading some readers.
Author Response
Dear reviewer,
We accepted all editorial-type changes and added information in accordance with comments â„–13.
A revised version of our manuscript was prepared accordingly.
Thank you very much for the comments and suggestions to improve the manuscript.
With kind regards,
Elena Gultyaeva